# Ex-Vivo Comparison of Torsional Stress on Nickel–Titanium Instruments Activated by Continuous Rotation or Adaptive Motion

**DOI:** 10.3390/ma13081900

**Published:** 2020-04-17

**Authors:** Joo Yeong Lee, Sang Won Kwak, Jung-Hong Ha, Hyeon-Cheol Kim

**Affiliations:** 1Department of Conservative Dentistry, Dental Research Institute, School of Dentistry, Pusan National University, Yangsan 50612, Korea; ljy8829@nate.com (J.Y.L.); endokwak@pusan.ac.kr (S.W.K.); 2Department of Conservative Dentistry, School of Dentistry, Kyungpook National University, Daegu 41940, Korea; endoking@knu.ac.kr

**Keywords:** adaptive motion, continuous rotation, torque generation, torsional stress, kinetics

## Abstract

This study aimed to evaluate the effect of adaptive motion applied to conventional nickel-titanium (NiTi) rotary instruments on torsional stress generation during shaping procedure. One hundred and twenty mesio-buccal canals of molars were randomly assigned to two groups according to the kinetics; adaptive motion (AD) and continuous rotation (CR). Each group was divided into four subgroups (n = 15) according to the NiTi instrument systems: HyFlex EDM, One Curve, Twisted File Adaptive, and ProTaper Next. A glide path was established with PathFile #1, for each file group being used with either of the kinetic movements. During the instrumentation with the designated motion and file system, the generated torque was measured via the control unit and acquisition module. Based on the acquired data, the maximum and total torque were calculated. The data were statistically analyzed using Kruskal–Wallis and Mann–Whitney tests at a significance level of 95%. The maximum and total torque generated by all instruments were significantly reduced by the adaptive motion (*p* < 0.05). In the CR group, HyFlex EDM generated the highest maximum and total stress. In the AD group, HyFlex EDM showed the highest maximum torsional stress, and One Curve showed the highest total torsional stress (*p* < 0.05). The TF Adaptive instrument with adaptive movement produced the lowest maximum and total torsional stress (*p* < 0.05). Under the conditions of this study, the use of adaptive motion would be useful to reduce the torsional stress of instrument and root dentin. The reduction of torsional stress through adaptive motion may enhance the durability of instruments and reduce the potential risk of dentinal cracks.

## 1. Introduction

Flexible and super-elastic properties of nickel–titanium (NiTi) endodontic instruments have brought significant enhancements in the outcome of endodontic treatment, allowing efficient cleaning and shaping of root canals [1,2,3,4]. Even though NiTi rotary instruments have been developed for several decades, there are still some limitations, which include the instrument separation and screw-in effect [3,5,6,7,8,9]. These can occur in the root canal during the root canal shaping procedure depending on several factors such as the skill of the operator, the shape of the root canal, and the nature of the instruments [3,5,6,9]. Recently, most newly-developed NiTi instruments have been made with heat-treated alloys, which increases the life in service of the endodontic instruments by recovering the super-elastic effect [3].

However, the breakage of instruments could occur even without any apparent signs of plastic deformation of the fracture site at various levels of stresses or strain [5,7,8,9]. Instrument separation inside the root canal may occur mainly via two mechanisms: torsional fracture and flexural fracture [10,11,12]. If the file rotates while it is stuck in the root canal, torsional fracture occurs [12,13]. When compressive and tensile forces are repeated on instruments in curved root canals, flexural stress also can cause the file to break [10,11]. To overcome these challenges and achieve a better performance, there have been a lot of efforts and researches in recent years. One of the most common ways is to reduce the stresses of the file by various changes in the design and alloys of instruments [14,15]. Instruments with an asymmetric cross-sectional area, variation of flute design, and increased flexibility via heat treatment can preserve the tooth structure without changing much of the original shape of the root canal [16,17,18].

Together with the alloy or heat treatment, movement kinetics has become a solution to increase the resistance of the file to separation [18,19]. Traditionally, a continuous rotary motion has been used for NiTi instruments before reciprocation movement was introduced. Regarding reciprocating movement, unequal bidirectional clockwise (CW)/counterclockwise (CCW) rotation has been proven to reduce instrumentation stress, and the concept of a single NiTi instrument to prepare the root canal has become utilized [20,21,22]. Unlike continuous rotation, stress can be relieved when the instrument rotates in a direction opposite to the twisting direction of the file [20,21,22]. Thus, the reciprocation NiTi instrument systems have become popular in recent years. However, there have been some controversies over the advantages and disadvantages of reciprocating motion in aspects of debris extrusion, canal transportation, and post-operative pain [23,24,25,26,27].

To utilize the advantages of both conventional continuous rotation and reciprocation, adaptive motion with the TF adaptive file has been proposed. The TF adaptive motion has a patented feedback algorithm, which changes the motion of the file based on instrumentation stress [8,18,26,28]. While there is less stress below a certain load on the file, according to the manufacturer, the movement is an interrupted clockwise rotation (CW 600–CCW 0°) that allows better cutting efficacy and debris removal. The more cutting of root canal dentin, the more stress it receives. At this point, the movement changes from clockwise rotation to reciprocation (CW 370–CCW 50°). These angles are not fixed but flexible with the stress the instrument receives. Thus, the manufacturer claims that the adaptive motion may select the optimal movement for each clinical situation with varying anatomical complexity. Previous studies have demonstrated that the adaptive motion would enhance the resistance to cyclical fatigue and torsional failure [8,18]. The adaptive motion rotates clockwise with a larger angle (370°) than the angles of reciprocal movement, which is 150° in Reciproc (VDW, Munich, Germany) and 180° in WaveOne (Dentsply Sirona, Ballaigues, Switzerland) systems. This system has non-fixed reciprocal movement with a larger angle of cutting directional rotation and would minimize the disadvantages of reciprocation without the deterioration of canal shaping efficacy [8,18,26,28]. Because motion kinetics do not affect the flexibility of the file, using a flexible file with adaptive motion may have a more synergistic effect in reducing stress during instrumentation.

There have been many studies on the canal shaping efficiency of adaptive motion, but there is a lack of studies comparing the torsional stress generation of instruments depending on movement kinetics. The aim of the present study was to compare the effect of motion type on the torsional stress generation during shaping procedure using various NiTi endodontic instrument systems.

## 2. Materials and Methods

### 2.1. Instrument Selection

In this study, the torsional stress generated during the canal shaping procedure was measured and compared across the four groups of NiTi rotary instruments: HyFlex EDM (HDM; Coltene-Whaledent, Allstetten, Switzerland), One Curve (OCV; Micro-Mega, Besançon, France), Twisted File Adaptive (TFA; Kerr Endodontics, Orange, CA, USA) size #25, and ProTaper Next (PTN; Dentsply Sirona) size X2. All the files were selected to have the same tip size of ISO #25 and 6% taper, except HDM, which has variable taper with apical 8%. However, these file systems have different geometries and are made of different NiTi alloy by heat treatment (Table 1). Before the experiment, any flaws or deformities of all instruments were checked by visual inspection, under a dental operating microscope (Leica S6D; Leica Microsystems, Wetzlar, Germany) at ×10 magnification.

### 2.2. Tooth Preparation

This study was approved by the Pusan National University Dental Hospital Institutional Review Board (PNUDH DRI-2018-02) and conducted in accordance with the Declaration of Helsinki. Previously extracted molars due to periodontal reasons were collected for this ex-vivo study. The teeth were stored in 4 °C distilled water until they were used in this study after the removal of periodontal soft tissue and calculus manually from the root surface. The exclusion criteria included teeth with an immature root apex, internal/external resorption, and any structural defect (caries, crack or fracture). Roots having more than a single apical foramen and root canal and root canal treatment were also excluded.

The teeth were checked radiographically to select for proper specimen. A standard radiograph was taken while each tooth was fixed with an EndoRay film holder (Dentsply Rinn, York, PA, USA). Canal curvature was calculated by using Schneider’s technique. On the radiograph, the angle between the line parallel to the canal axis in the coronal third and a line drawn from the apical foramen to intersect the point where the first line left the long axis of the canal was measured. According to the criteria, one hundred and fifty mesiobuccal canals with a moderate angle of root canal curvature (0–20°) were selected.

After the access cavity preparation of the selected teeth, gross pulpal tissue was extirpated using #20 H-file (Mani, Utsunomiya, Japan). The coronal part was resected with a high speed diamond bur (TR-13; Mani, Utsunomiya, Japan) at the level of the cemento-enamel junction. Root canal patency was established with #10 K-file (Mani, Utsunomiya, Japan). The working length was decided visually using #10 K-file. The file was inserted into the root canal until the tip was identified at the root apex and the working length was recorded as 0.5mm shorter than the measured length. After the glide path preparation with PathFile #1 (Dentsply Sirona, Ballaigues, Switzerland) at the suggested setting (300 rpm, 2 N·cm), the coronal portion was trimmed to standardize the working length to 15 mm. Then, the working length was reconfirmed with standard film radiography. Prepared teeth were preserved in 100% humidity.

### 2.3. Torsional Load Measurement

Each canal was randomly assigned to one of two groups according to the kinetics: the continuous rotation (CR) group and the adaptive motion (AD) group (n = 60). Each group was divided into four subgroups according to the NiTi rotary file: HDM, OCV, TFA, and PTN.

Preliminary tests were conducted to calibrate and determine the length of the pecking depth and the total number of pecking motions to reach the working length. Each root canal shaping procedure was carried out according to the following protocol: nine pecking movements were conducted to reach the working length (15 mm) and two more strokes were added at the working length. After every three strokes, debris was removed using compression air, and the root canal was irrigated with 2.5% NaOCl. The total instrumentation time was measured to control the test condition standardized to make it at around for 23 s. All instruments were operated with a same endodontic motor (Elements Motor, Kerr Endodontics).

In the CR group, the motor was set at the speed and torque values recommended by the manufacturer of each file. TFA was used in accordance with the operating protocol of Twisted File, a product of the same manufacturer. In the AD group, the motor was set to TF adaptive mode. To minimize operative errors, an endodontist with ten years of experience performed the canal shaping procedures. During the shaping procedure, the torque generated was extracted from the motor and recorded at a rate of 100 Hz by using a data acquisition module and dedicated software programed for a custom device (AEndoS-*P*; DMJ system, Busan, Korea) throughout the shaping procedure (Figure 1).

Based on the acquired data, the instrumentation time (X axis) and torsional load (Y axis) were plotted using Origin v6.0 Professional software (Microcal Software Inc, Northampton, MA, USA). The maximum value of the plot was extracted as maximum torque, and the integration of the plot of varying torque was recorded as total torque.

### 2.4. Statistical Analysis

The acquired data were evaluated for the normality of distribution using the Shapiro–Wilk test. Due to the abnormal data distributions, Kruskal–Wallis and Mann–Whitney tests were used to determine the statistical significances between movement kinetics and the type of NiTi instruments. The statistical analysis was conducted using the SPSS software (Version 22.0; IBM, Armonk, NY, USA) and the significance level was set at 95%.

## 3. Results

### 3.1. Total Torsional Load

The generated total torques by all instruments were significantly reduced by the adaptive motion (Figure 2 and Table 2, *p* < 0.05). The average of the total torsional load was significantly higher in the CR group than in the AD group (*p* < 0.05).

In the CR group, the total torque was the highest in HDM and the lowest in TFA. There was no significant difference between the OCV and PTN. In the AD group, OCV showed the highest total torsional load and decreased in the following order; HDM, PTN, and TFA. Irrespective of the operating system, TFA generated the lowest torque.

The difference in total torsional load in terms of motion kinetics was biggest in the HDM subgroup and smallest in the TFA group. TFA instruments with adaptive movement produced the lowest total torsional load.

### 3.2. Maximum Torque

The torsional load generated during the root canal shaping process was increased gradually and then decreased after reaching the working length in all groups. The generated maximum torques by all instruments were significantly reduced by the adaptive motion (Figure 2 and Table 2, *p* < 0.05). The CR group presented a significantly higher maximum torsional load than the AD group regardless of the type of instrument, except for TFA (*p* < 0.05).

In the CR group, HDM generated the highest maximum torque, followed by OCV, PTN, and TFA showed the least. In the AD group, HDM generated the highest maximum torsional load, followed by OCV, PTN, and TFA.

The difference in maximum torque in terms of motion kinetics was biggest in the HDM subgroup and smallest in the TFA group. TFA instruments with adaptive movement produced the lowest maximum torsional load.

## 4. Discussion

Torque is necessary for cutting and removing infected dentin during instrumentation, and torsional load is caused naturally by friction between the instrument and the root canal surface during the root canal shaping procedure [12,13]. When the instrument is interrupted in the root canal, the frictional force between the instrument and the root canal wall creates a torsional load build-up in the instrument. Kim et al. [29] reported that the internal residual stress was found even after the instrument was removed from the root canal. The presence of internal residual stresses may implicate that a certain amount of cold work in the microstructure of instruments can occur, which could make the instrument less flexible or more brittle [29,30]. If the torsional stress was generated immediately after exceeding the elastic limit, the instrument may be plastically deformed and fractured. Reaction force on the root dentin is also concentrated as a result of being in contact with the instrument [29]. Many studies reported that the stress generated during root canal enlargement may cause instrument breakage as well as dentinal damage or cracks [10,21,31].

Torque generation during instrumentation is affected by various factors [12,13,14,16,17,18]: the anatomy of the root canal, the rotational speed and torque setting of the endodontic motor, the size or taper of the instrument, and the defect of instrument surface influence on the generation of torsional load. Usually, the instrument is subjected to a higher torsional load at the initial stage of root canal enlargement [5,32]. Especially in narrow root canals, bigger torsional load would be generated to maintain constant rotation speed [5,32]. Meanwhile, Kim et al. [29] reported that higher stresses in the root during instrumentation can be expected to increase dentinal defects. Dane et al. [33] reported that instrumentation of the root canal used at a high torque setting might cause more crack formation in the root canal dentin than used at a low torque setting. The higher setting of torque limits permits higher load to occur, hence, more stress is transferred to the dentin and the possibility of a dentinal crack would be higher [33]. All the torques for root canal shaping are directly related to the potential risk of damage to the root dentin as well as instrument fracture.

To control this torsional load generation, clinicians may use more flexible instruments or a lower torque setting during usage of the NiTi instruments [14,33,34]. The kinematic movements are very important criteria to determine the stress generation during the file movement. Reciprocating movements were reported to reduce torsional stress by periodic change of the rotation direction, with very small angles of rotation ranging from 30° to 150° [8,10,20,21]. This reciprocating movement was reported to reduce the screw-in forces because it does not rotate continuously while the conventional instruments rotate and have screw-in forces during rotation [6,8]. Adaptive motion as another kinetic movement may have other advantages beyond the reciprocating movement and the conventional continuous rotation. Adaptive motion would help to reduce cyclical fatigue by switching from a rotary to a reciprocation motion [8,10,18]. Thus, the lifespan of the files would be expanding. Secondly, the adaptive motion lessens the twisting load on the file, which allows the instrument to avoid reaching a stress level that might normally unwind or fracture the file [8,10,18].

Unlike typical torque control, adaptive motion helps clinicians reduce the risk of file fracture and improve clinical torque control [10,20]. Therefore, this study was performed to verify the actual effects on the torsional load and other potential changes to expand the application on the other NiTi file systems.

The results in this study showed that the adaptive motion used for each heat-treated instrument systems generally showed a good performance to reduce the maximum torque generation as well as the total torsional load during instrumentation procedures. Recently, the coincided results were published by comparing continuous rotation and adaptive motion using resin-simulated root canals [18].

Torsional load was generated for every single pecking motion, and the torque decreased considerably after the instrument reached the working length. In this study, the maximum torque occurred almost immediately before reaching the working length. Kwak et al. [32] reported that the torque was affected by the canal anatomy and the maximum torque was generated when the file passed through the canal curvature rather than the straight part of the root canal. Because teeth with a moderate curvature below 20° were included in the present study, the tendency of the greatest torque value was not clearly matched with that in passing through the curvature section. The irregularity of natural teeth would also result in a difference in the timing of maximum torque generation.

The torque generation is related to the stiffness of the instruments that is directly related to the elasticity [13,14,31,35]. The elasticity of the instrument would be decided from the alloy property and geometries. The instruments in this study are made of R-phase, M-wire, C-wire, and CM-wire. Among these alloys, M-wire and R-phase may have less martensitic phase and higher stiffness and consequent stress generation than C-wire and CM-wire [14,15,35,36,37]. However, based on the results, the geometric characteristics, especially the taper and sizes, seemed to have a higher effect on stress generation [16,17].

The tested instruments in this study had the same D0 size. Among the five file systems used in this study, while PTN, OCV, and TFA have a 6% taper, HDM has an 8% taper at the apical area and a variable taper at the coronal area. The progressive changing taper of HDM may reduce the volumetric dimension and diameter of the file at the mid-body area and make them thicker in coronal area. However, the apical tip portion of HDM has an 8% taper and a bigger volume and diameter than other 6% files. For that reason, the HDM seems to have higher torque generation than other files [16,36].

Extracted human molars were used to reproduce the conditions as close as possible to the clinical situation. Since natural teeth have various anatomical forms and structures, it was difficult to standardize initial conditions. For as much standardization of specimens as possible, we selected mature roots with similar shapes and calculated the curvature of the canal for exclusion of severely-curved root canals. Working length measuring and glide path establishment were followed by the same procedure in all specimens to set similar canal conditions. Although the manufacturer recommended the dedicated glide path preparation instrument before using the shaping files, the single brand of PathFile #1 was used for all groups to make a standardized condition, at least, of root canals, and to maintain the smaller pre-operative root canal size, which is advantageous in obtaining torque generation values by the NiTi file systems.

A custom device and program used in this study could extract the torque information generated in an instrument while operating in the canal in real time from the endodontic motor. This device allowed quantitative measurement of the amount of torque generated in the instrument and the root canal during the operation.

Collectively, from the present study, maximum torque and total torsional load were decreased when operating with adaptive motion. Although the reduced amount of torsional load depending on the type of movement kinetics varies according to the type of instrument, it could be speculated that adaptive motion may reduce the concentration of torsional stress on the instrument and the root dentin. Adaptive motion alters the movement of the file based on the applied load according to the patented feedback algorithm. The motor continuously rotates until high torque is applied. As the torque increases, the motor responds and starts reciprocation. Reciprocation with a larger clockwise angle has benefits: the removal of dentin and the decrease of torsional load [8,10,18,26,28].

Recently, the new generation of rotary instruments has drawn much attention from clinicians for its great improvement in the shaping ability and preservation of the natural anatomy with a minimally invasive procedure. Various instruments with different heat treatments and specially-designed instruments may be other candidates for adaptive usage and valuable for future studies.

The main limitation of the present study is the absence of information about pulpal conditions at the time of tooth extraction. Factors such as the difference between maxillary molar and mandibular molar and the age of the tooth donor can affect the outcome. Experimental conditions with more strict control of tooth conditions are recommended for further in vivo studies.

## 5. Conclusions

Under the limitations of this study, it could be concluded that the use of adaptive motion for endodontic rotary instruments would be useful to reduce the torsional stress of the instrument and root dentin. The reduction of torsional stress through adaptive motion could enhance the durability of instruments and decrease dentinal cracks.

## Figures and Tables

**Figure 1 materials-13-01900-f001:**
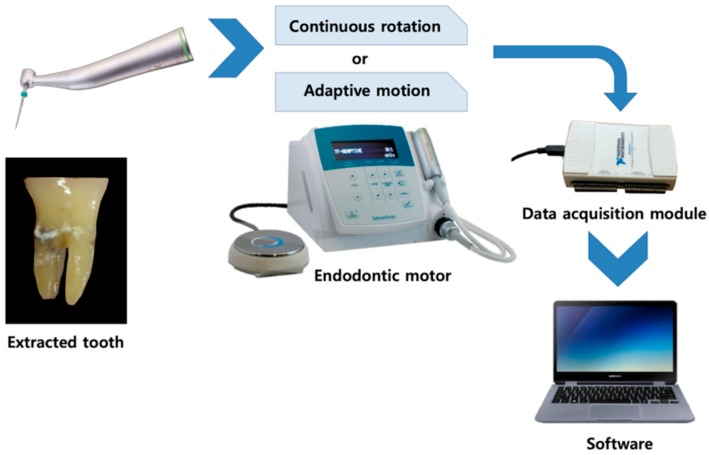
Diagram for test setup. Extracted teeth were used and two modes of kinetics of file movement were applied. The torsional load during instrumentation was recorded using a custom device and program.

**Figure 2 materials-13-01900-f002:**
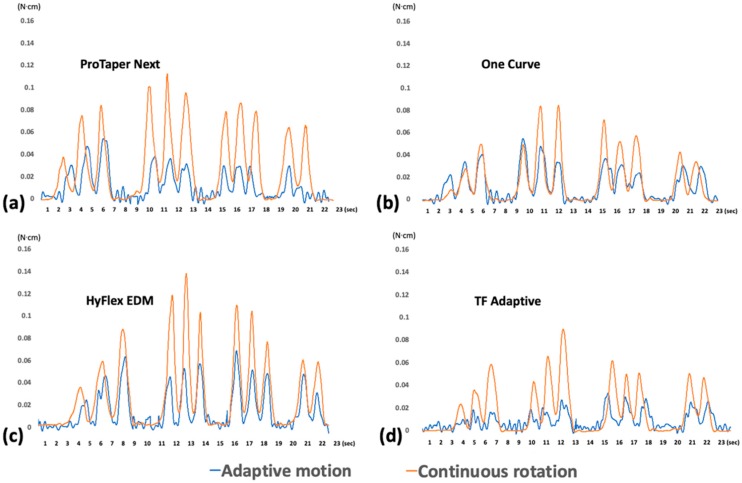
Representative torsional load generation during the adaptive motion (blue) and continuous rotation (orange) for One Curve system: (**a**) ProTaper Next; (**b**) One Curve; (**c**) HyFlex EDM; (**d**) TF Adaptive.

**Table 1 materials-13-01900-t001:** Specification of tested files and recommended rotational speed and torque.

File	Manufacturer	Specification			Recommended Rotation Speed (rpm) and Torque (gcm)	
		Cross-Section	Taper	Wire	rpm	gcm
ProTaper Next	Dentsply Sirona, Ballaigues, Switzerland	off-centered rectangle	Apical .06 and variable	M-wire	300	200
One Curve	Micro-Mega, Besançon, France	Variable	.06	C-wire	300	250
HyFlex EDM	Coltene-Whaledent, Allstetten, Switzerland	Variable	Apical .08 and variable	CM-wire	400	250
Twisted File Adaptive	Kerr Endodontics, Orange, CA, USA	Triangle	.06	R-phase	500	200

**Table 2 materials-13-01900-t002:** Maximum torque (N·cm) and total torque (N·cm) according to the kinetic motion and file systems.

File	Maximum Torque (N·cm) *		Total Torsional Load (N·cm) *	
	Continuous Rotation	Adaptive Motion	Continuous Rotation	Adaptive Motion
ProTaper Next	0.092 ± 0.036 ab	0.042 ± 0.007 b	0.523 ± 0.151 b	0.214 ± 0.051 ab
One Curve	0.106 ± 0.045 bc	0.046 ± 0.011 b	0.471 ± 0.132 b	0.230 ± 0.049 bc
HyFlex EDM	0.184 ± 0.125 d	0.059 ± 0.012 c	0.741 ± 0.439 c	0.218 ± 0.066 ab
TF Adaptive	0.048 ± 0.025 a	0.033 ± 0.004 a	0.185 ± 0.045 a	0.172 ± 0.019 a

* The maximum torque and total torsional load was influenced by the movement kinetics (*p* < 0.05). a,b,c,d: The different letters indicate significant differences between file groups (*p* < 0.05).

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
