# Peer review of "Ex-Vivo Comparison of Torsional Stress on Nickel–Titanium Instruments Activated by Continuous Rotation or Adaptive Motion"

_materials, 2020, doi:10.3390/ma13081900_

Round 1
Reviewer 1 Report
Dear authors
Thank you for the submission of the present manuscript. The results are interesting and useful for application in clinical practice. After a careful review of the manuscript, I only have two suggestions/corrections that you should consider revising.
First, I believe that you should delete the phrase "decrease of dentinal crack" from Abstract and conclusions because it is an assumption and does not supported by your results.
Second, you should give an explanation in the manuscript why your data had abnormal distribution and you had to perform non parametric tests. Was it due to small sample size or it was due to another reason?
Author Response
First, I believe that you should delete the phrase "decrease of dentinal crack" from Abstract and conclusions because it is an assumption and does not supported by your results.
Authors’ response: Thank you for this notice. We have revised the last sentence under the condition of the conducted study.
Second, you should give an explanation in the manuscript why your data had abnormal distribution and you had to perform non parametric tests. Was it due to small sample size or it was due to another reason?
Authors’ response: Thank you for this comment. This study was conducted with the extracted teeth so that the root canals have varying size with the natural anatomy. Thus the attained values showed to have big deviation and therefore we should do the statistic analysis with nonparametric methods.
Reviewer 2 Report
My background is engineering and fracture and fatigue of metals and I have works in NiTi failures; the manuscript is much more oriented to Dentistry area than engineering therefore is this adequate for Materials journal?
Anyway i think, that from my pot of view (engineering) the following items must be explained:
-which mechanical properties have the different files used? are similar or different?
-which methodology was used to measure the torque? with sensors on the file or in the motor and how it was calibrated
-in engineering units ar N and mm, why the use of N.cm in the torque? Please correct
-the discussion section is much more oriented to dentistry than for materials research
Author Response
My background is engineering and fracture and fatigue of metals and I have works in NiTi failures; the manuscript is much more oriented to Dentistry area than engineering therefore is this adequate for Materials journal?
Authors’ response: This manuscript was submitted to the special issue of “Contemporary Endodontic Material” and the Endodontics is a branch of Dentistry.
Anyway i think, that from my pot of view (engineering) the following items must be explained:
-which mechanical properties have the different files used? are similar or different?
Authors’ response: Thank you for this notice. We have added further description about the files used in this study with their properties.
-which methodology was used to measure the torque? with sensors on the file or in the motor and how it was calibrated
Authors’ response: The endodontic motor moves under a set condition by the control panel and the motor and control unit measure the torque generated during the file rotation within the root canal. The torque has been extracted from the control unit to the computer via the acquisition module and it could be calibrated with the dedicated software in the computer.
-in engineering units ar N and mm, why the use of N.cm in the torque? Please correct
Authors’ response: The endodontic motor uses the unit Ncm due to the low value (not exactly but endodontics use this unit). Thank you.
-the discussion section is much more oriented to dentistry than for materials research
Authors’ response: Yes, it is. Because this article is for dental field especially, endodontics. Thank you.
Reviewer 3 Report
The article deals with an interesting topic, that is the torsional stress of NiTi instruments induced by different kinematics.
However, some changes are required and some foundamental explanations concerning the protocol are needed.
Abstract:
I recommend to verify the abstract structure required by the journal.
Please remove "the Introduction" after Abstract
I suggest to use "kinematics", instead of "kinetics"
"during final instrumentation": in the M&M section, you state that measurements are done during the shaping procedure. Please explain or correct.
"the torque was measured." Pleasy clarify how
M&M:
Please specify the variable taper of Hyflex EDM (Ref. Pirani et al. 2015). The taper of these files is one of the main bias of the research: different files with different taper are compared. This inevitably creates differences between torsional stress and for this reason authors should give explanation about this limit.
Please specify speed and torque setting for each file.
Please explain when and how the torsional stress was measured.
The used protocol is #13 (Pathfile 1) for glidepath creation and than shaping files. This significantly differs from the majority of manufacturers recommendation (Proglider use, Glide path file Hyflex use....) that suggest the use of files with increased diameters. A 25 diameter file is used after a 13 diameter and this greatly stresses the file tip. Please discuss this aspect.
The device used to record the stress should be decribed in depth.
Results:
Fig.2 caption. Please revise it (for OneCurve instruments)
Discussion:
Authors write about "conventional instruments". Files with heat treated alloys are compared so it would be better to remove conventional NiTi.
You state that PTN files have a constant taper. I think this is not correct: the taper is increasing 6%-7% and than decreased in the coronal portion.
OC have a constant taper. Please cite Ertugrul 2019.
Pag. 7: you discuss about the metallographic features of NiTi tested files. I suggest to add references (Shen et al. 2013; Iacono et al. 2017) for CM wire, m-wire, r-phase. What abput C-wire? do you have any references for this?
EDM files generate more stress than other files. This is due to the increased taper of this system. Authors should discuss this great bias.
Author Response
I recommend to verify the abstract structure required by the journal.
Authors’ response: Thank you for this notice. We have checked the guideline and corrected accordingly.
Please remove "the Introduction" after Abstract
Authors’ response: Thank you again for this notice.
I suggest to use "kinematics", instead of "kinetics"
Authors’ response: Thank you. We have accepted this suggestion and applied to the manuscript..
"during final instrumentation": in the M&M section, you state that measurements are done during the shaping procedure. Please explain or correct.
Authors’ response: Thank you for this notice. The “final” is wrong word. We have corrected the abstract section.
"the torque was measured." Pleasy clarify how
Authors’ response: Thank you. Some words added although the abstract has the limitation of words number.
M&M:
Please specify the variable taper of Hyflex EDM (Ref. Pirani et al. 2015). The taper of these files is one of the main bias of the research: different files with different taper are compared. This inevitably creates differences between torsional stress and for this reason authors should give explanation about this limit.
Authors’ response: Thank you for this notice. From the view point of clinic, we need to compare the different brands of instruments although they are different from each other. The main purpose of this study is to compare the kinetics between traditional rotation and adaptive motion. The second variable was file brand. Thus the figures in the results present the comparison within the same instrument.
Please specify speed and torque setting for each file.
Authors’ response: Thank you. It is added in the new table 1.
Please explain when and how the torsional stress was measured.
Authors’ response: Thank you for this notice. Actually, we did not measure or calculate the stresses. The TORQUE was measured by endodontic motor during the shaping procedure. The motor/handpiece was connected to the control unit which can measure the torque via handpiece and we took out the data vales to the analyzing computer. We have revised the nomenclature of stresses to torque correctly.
The used protocol is #13 (Pathfile 1) for glidepath creation and than shaping files. This significantly differs from the majority of manufacturers recommendation (Proglider use, Glide path file Hyflex use....) that suggest the use of files with increased diameters. A 25 diameter file is used after a 13 diameter and this greatly stresses the file tip. Please discuss this aspect.
Authors’ response: Thank you for this comment. The torques are affected by original canal lumen size, definitely. If the canal is prepared with various tapered or sized glide path NiTi file, the pre-operative canal volume might vary and this could affect the results. Thus, it is hard to collect the accurate data and see the effect from the KINETICS and NiTi file systems. The reason why we use PathFile 1 (#13) instead of other larger files were to make a MINIMAL standardized condition of root canals and maintaining the smaller pre-operative root canal size is advantageous in obtaining torque generation values by preparation from each NiTi file systems. We added it in the discussion section.
The device used to record the stress should be decribed in depth.
Authors’ response: Thank you. The device AEndoS is originally designed to record all the torque and pressure data generated during procedure for the in vitro test. The program to control the device can measure the data as well as control the speed, angle and torque of the devices.
This study was done by the software to analyze the torque data from the handpiece and control unit. The data extracted from the control unit was transferred through an acquisition module for coupling customized program. When the motor starts, the torque is generated at any time and the computer is able to collect all the torque data during the procedure in real-time. We have added further in the methods.
Results:
Fig.2 caption. Please revise it (for OneCurve instruments)
Authors’ response: The One Curve is correct. Manufacturer does not use OneCurve. Thank you.
Discussion:
Authors write about "conventional instruments". Files with heat treated alloys are compared so it would be better to remove conventional NiTi.
Authors’ response: Thank you. The manuscript was revised.
You state that PTN files have a constant taper. I think this is not correct: the taper is increasing 6%-7% and than decreased in the coronal portion. OC have a constant taper. Please cite Ertugrul 2019.
Authors’ response: Thank you. Yes, PTN has various tapers (6% taper at 3 mm from the tip, followed by a 7% taper up to 9 mm, and then the taper decreased up to 16 mm). We intended that PTN, OCV, and TFA have 3 mm constant taper at the file tip, but the manuscript was revised because there could be misunderstanding.
Pag. 7: you discuss about the metallographic features of NiTi tested files. I suggest to add references (Shen et al. 2013; Iacono et al. 2017) for CM wire, m-wire, r-phase. What abput C-wire? do you have any references for this?
Authors’ response: Thank you. We added more references. C-wire is almost same or similar to the Gold wire but much softer than this. It is obvious that the C-wire is also heat treated but there is lack of information about it. Anyhow we have added another reference for C-wire.
EDM files generate more stress than other files. This is due to the increased taper of this system. Authors should discuss this great bias.
Authors’ response: As we mentioned in manuscript, the torque generation is largely affected by the contact between canal wall and file system. EDM file has lager taper than another tested files. Furthermore, due to EDM files’ cross sectional design, some part of the file shows a surface contact with the canal wall rather than a point contact. These features may affect the increased amount of generated torque. These comments are discussed in the original manuscript and revised. Thank you.
Round 2
Reviewer 2 Report
The authors did not answer to my comments; anyway since the paper is completely out-of-scope of engineering and matérias, but only on endodontics, I leave the decision to the others reviewers
Author Response
The authors did not answer to my comments; anyway since the paper is completely out-of-scope of engineering and matérias, but only on endodontics, I leave the decision to the others reviewers.
Authors' response: We have responded on this issue as below at the first round of review and revision.
This manuscript was submitted to the special issue of “Contemporary Endodontic Material” and the Endodontics is a branch of Dentistry.
So the journal "materials" can publish this topic in this special issue.
Thank you.
Reviewer 3 Report
The article has been corrected and improved as suggested.
I only recommend to the authors to revise the last two references because some mistakes are present.
Author Response
I only recommend to the authors to revise the last two references because some mistakes are present.
Authors' response: Thank you. We have checked it and corrected one for the reference list.